# Knowledge Distillation for Traversable Region Detection of LiDAR Scan in Off-Road Environments

**DOI:** 10.3390/s24010079

**Published:** 2023-12-22

**Authors:** Nahyeong Kim, Jhonghyun An

**Affiliations:** School of Computing, Gachon University, Seongnam-si 1332, Republic of Korea; skgud99@gachon.ac.kr

**Keywords:** knowledge distillation, off-road, LiDAR point cloud, self-driving, point cloud projection, range image

## Abstract

In this study, we propose a knowledge distillation (KD) method for segmenting off-road environment range images. Unlike urban environments, off-road terrains are irregular and pose a higher risk to hardware. Therefore, off-road self-driving systems are required to be computationally efficient. We used LiDAR point cloud range images to address this challenge. The three-dimensional (3D) point cloud data, which are rich in detail, require substantial computational resources. To mitigate this problem, we employ a projection method to convert the image into a two-dimensional (2D) image format using depth information. Our soft label-based knowledge distillation (SLKD) effectively transfers knowledge from a large teacher network to a lightweight student network. We evaluated SLKD using the RELLIS-3D off-road environment dataset, measuring the performance with respect to the mean intersection of union (mIoU) and GPU floating point operations per second (GFLOPS). The experimental results demonstrate that SLKD achieves a favorable trade-off between mIoU and GFLOPS when comparing teacher and student networks. This approach shows promise for enabling efficient off-road autonomous systems with reduced computational costs.

## 1. Introduction

With the increasing use of autonomous systems in vehicles, detection in the real world has become a crucial task [1]. Among the various detection tasks, identifying a traversable region to ensure safe driving in changing conditions accounts for a significant proportion of the system’s workload. This task can be divided into two parts: performance in urban and off-road environments [2,3]. Urban environments are equipped with signals such as traffic lights, road lanes, and kerbs on pavements. In contrast, off-road environments include irregular driving conditions and obstacles such as trees, stones, and puddles. In addition, off-road environments lack clear boundaries, such as lanes and kerbs; therefore, there is a lack of detail about traversable regions in off-road environments compared with urban areas. Therefore, it is advantageous to use three-dimensional (3D) spatial information to recognize off-road environments. LiDAR sensors are a suitable choice for this. Existing research on determining driving areas has primarily been conducted through segmentation based on images [4,5,6]. However, because an image contains only two-dimensional (2D) information, it is difficult to determine the specific shape, space, and depth of the real world [7]. This problem is being addressed by the emergence of 3D point cloud data, leading to more effective solutions.

As 3D point cloud technology draws attention and evolves, various LiDAR-based studies have flourished. Notably, areas such as SLAM [8], object detection [9,10], and semantic segmentation [11] have seen significant developments. Through these studies, tasks such as terrain analysis, obstacle avoidance, and environmental monitoring in autonomous driving can be accomplished. Substantial research is needed to further the progress of autonomous driving in light of these advancements. However, 3D point cloud data comprise higher dimensions than images, which are larger, more complex, and irregular. This means that point cloud data are complex to process and require a higher computational cost, meaning that the segmentation of a point cloud is challenging [12,13,14]. Therefore, a method has been devised to project a 3D point cloud onto a 2D image and use it as a range image [15]. Unlike 3D point clouds, range images can be processed as images. In addition, because each point is projected with a pixel, it contains the distance and depth information in the form of a 2D image. This means that the point cloud can be used with less computational power, but with richer information than the 2D image.

On an unpaved road, the overall driving area is rough and dangerous; therefore, the risk of damage to hardware is higher than that on a paved road. In the event of hardware damage, emergency services and repair centers are well established in urban areas. However, off-road environments lack sufficient infrastructure compared to the level of risk, as seen in urban areas. In addition, because it is used for purposes such as reconnaissance missions, agriculture, and mining industries, it is challenging to schedule frequent maintenance due to its extended operational time [16,17,18]. It is burdensome to mount high-priced, high spec equipment for hardware with these issues that is used in off-road environments; thus, conducting studies to reduce computational cost in such environments is crucial [18]. Therefore, network compression is attracting attention for reducing computational costs when processing point clouds, images, and various other forms of data in deep learning applications [19]. Many studies have been conducted on knowledge distillation (KD), which divides the knowledge of a large, cumbersome teacher network into lightweight student networks [20,21,22]. In this study, we proposed a KD method for segmenting range images in off-road environments. The experiments were performed using the off-road environment dataset RELLIS-3D. The proposed soft-label knowledge distillation (SLKD) performs KD on off-road range image segmentation, obtaining both depth information and segmentation output at a low computational cost. In summary, the main contributions of this paper are summarized as follows:We propose soft label knowledge distillation (SLKD) for off-road range view images. To the best of the authors’ knowledge, SLKD is the first method that distills knowledge for semantic segmentation of 3D point cloud range image in off-road view;We conduct experiments on the RELLIS-3D benchmark and demonstrated that SLKD achieves a significant improvement in mIoU performance when using the same computational costs;We evaluated the distillation performance with several student encoders, demonstrating the robustness of SLKD.

## 2. Related Works

### 2.1. Point Cloud Processing

Point cloud data have three main characteristics. First, as shown in Figure 1a, depending on the sensor’s measurement position, the target object is measured densely in some areas and sparsely in others. This is called an irregularity. Second, as shown in Figure 1b, there exists a feature that does not follow any structural pattern or rule, and the distance between all points is not constant; this is called an unstructured feature. Third, as shown in Figure 1c, all points are not aligned and, as a set, the point clouds are permutation-invariant. This characteristic is expressed as an unordered point cloud. Various methods have been devised to address irregular, unstructured, and unordered point clouds.

PointNet [12] was the first method designed to directly process a point cloud. Each layer of the network processes all the points identically or independently. Max pooling is used to learn the criteria for choosing interesting informative points and to encode the basis for these criteria. Next, the results of the max pooling were collected using FC layers and converted into a global descriptor. This descriptor is used for entry shape or per-point labels. However, PointNet does not fully reflect the information in the metric space in which a point exists when learning the local structure. To overcome this problem, PointNet++ [23] was developed. PointNet++ introduced hierarchical natural networks that nested the partitioning of the input point set and recursively applied PointNet to this structure. Thus, local features can be learned with improved contextual scaling. With this process, PointNet++ enables the capture of both local and global features of the point cloud, making it more effective for various applications, such as 3D object recognition.

Instead of a raw point cloud, the voxel-based method uses points in the form of a voxel that divides the 3D space into a certain grid. VoxelNet [24] creates features of voxels using the points in each voxel through a voxel feature encoding (VFE) layer. Voxels that obtain features integrate the local voxel features through 3D convolution. The point cloud is then utilized as an input to the region proposal network. SqueezeSeg [15] proposed a method for reorganizing a point cloud into a spherical range image for use as an image. The label map for each point was output by utilizing the reconfigured point cloud as the input for the CNN. The label map was then reinforced using a conditional random field (CRF) implemented as a recurrent layer. Similarly, SalsaNext [25] processes the full 3D LiDAR point cloud by projecting it onto a 2D range image. SalsaNext is the next version of SalsaNet [26] and has an encoder–decoder architecture, where the encoder unit has a set of ResNet [27] blocks and the decoder part combines up-sampled features from the residual blocks. A pixel-shuffle layer was added to the decoder, whereas SalsaNet’s ResNet encoder blocks were replaced with a new residual dilated convolution stack. Methods for applying the 3D point cloud range image as an input to the U-Net [28] structure have also been studied [29].

### 2.2. Traversable Region Detection

A method to determine the traversable region during self-driving was developed for semantic segmentation. Semantic segmentation is a computer vision task that involves precise labeling of individual pixels in an image or video to determine the specific class or category to which each pixel belongs. Applying semantic segmentation to a road environment can help us to understand the class to which each pixel belongs, such as roads, sidewalks, vehicles, and pedestrians. Similarly, segmentation in the point cloud can be considered a task to determine the class to which each point belongs. The gated-shape CNN (GSCNN) [30] utilizes two streams: a regular stream and a shape stream. Verifying the network using the Cityscapes dataset [31] demonstrated sophisticated traversable region detection in an urban environment. Xie et al. [32] proposed a semantic segmentation framework that unified transformers using lightweight multilayer perception (MLP) decoders. The proposed SegFormer [32] showed robustness in the evaluation of a cityscape dataset and identified drivable areas. Yan et al. [33] proposed a fusion-based approach that uses both a camera and LiDAR for semantic segmentation. The performance of the proposed 2D priors-assisted semantic segmentation (2DPASS) was verified with semantic KITTI [34], a LiDAR point cloud dataset for urban environments. Cylinder 3D [35] also effectively performed 3D urban view semantic segmentation using the semantic KITTI. Although studies on determining traversable regions for urban environments are being actively conducted, studies on off-road views are lacking. For urban scenes, there are various datasets such as Cityscapes [31], Semantic KITTI [34], Nuscenes [36], Waymo [37], and BDD100K [38]. In contrast, there are fewer datasets available for off-road scenes. Here, the three main datasets for off-road scenes are RELLIS-3D [39], RUGD [40], and Deep-Scene [41].

Off-road self-driving can be used in a variety of fields; however, compared to urban areas, it is difficult to collect data owing to the unstable and rough driving environment. Therefore, this field needs more research. This study was conducted to determine the traversable regions in an off-road environment rather than in an urban scene.

### 2.3. Knowledge Distillation

Recently, various KD techniques have been proposed to reduce the computational cost of deep learning [42,43,44,45]. The goal of KD is to transfer the knowledge of a large, cumbersome teacher network to a lightweight student network. Many types of KD have been studied, and knowledge transfer can be broadly divided into response-, feature-, and relationship-based.

Response-based knowledge transfer generally refers to the response of the last output layer of a teacher network. This can be considered as directly mimicking the final prediction of the teacher network. Hinton et al. [22] proposed logit distillation using response-based knowledge. The proposed vanilla KD reduces the performance gap between the two models by distilling knowledge and allows for more efficient use of the student network. Feature-based knowledge uses the characteristics of deep learning with multiple layers. In addition to the response of the last output layer used in the response-based knowledge, a feature map, which is the output of the middle layer, is used as feature-based knowledge. The feature map of the teacher network can be used as knowledge of the student network. Romero et al. [46] proposed a method for directly matching a teacher’s feature activation to that of a student. The proposed FitNets performs even better than the teacher network using fewer parameters. Response-based and feature-based knowledge utilize the specific layer’s output of the teacher model as knowledge. Relation-based knowledge explores the different layers of data samples. Yim et al. [42] proposed a method to distill knowledge by comparing the relationships between the features of teacher and student networks. The proposed flow-of-solution process (FSP) summarizes the relationships among feature map fairs. Using the proposed method, they achieved faster optimization.

The SLKD proposed performs logit distillation using response-based knowledge. Using our SLKD, we can effectively transfer a teacher’s knowledge to a student in the range image of an off-road environment. To the best of our knowledge, SLKD is the first study that applies KD-to-LiDAR point cloud range images in off-road environments.

## 3. Method

### 3.1. RELLIS-3D

Many self-driving datasets exist, including the Semantic KITTI, BDD100K, Cityscapes, Waymo, and nuScenes. However, the aforementioned datasets and other public datasets primarily represent urban environments. Off-road environment datasets are unusual compared to urban autonomy datasets because of the difficulties in collecting irregular driving environments. RELLIS-3D, RUGD, and Deep-Scene datasets include off-road environments. Among these, RELLIS-3D is the only one that includes LiDAR scans, unlike RUGD and Deep-Scene. The RELLIS-3D dataset was deployed for robust and safe semantic scene understanding in off-road environments. This dataset was collected at the Texas A&M university campus and includes RGB camera images, LiDAR point clouds, etc. It contains annotations for 13,556 LiDAR scans and 6235 images and consists of a total of 20 classes such as dirt, grass, tree, and pole. However, in RELLIS-3D’s LiDAR scan, as shown in Figure 2, grass, trees, and bushes account for approximately 80% of the total point labels, and the distribution by point class is unbalanced. Various methods exist for resolving class imbalances in a dataset, such as undersampling [47], SMOTE [47], and multiclass classification [48]. Unlike in a city, distinguishing between traffic lights and signs during off-road autonomous driving is meaningless.

Therefore, Viswanath et al. [48] proposed a multiclass classification that integrates objects such as people, pillars, and trees into one class called obstacles, and also integrates other classes into a total of four classes, including traversable and non-traversable areas and the sky. The proposed method performed pooling for an image class of RELLIS-3D. In addition, when determining traversable regions, class pooling can improve the accuracy by correctly judging a previously incorrectly determined class as a pooled class [49]. Therefore, in this study, pooling [48] of the point classes of this dataset was performed, as shown in Table 1, to resolve the class imbalance and obtain efficient segmentation performance.

### 3.2. LiDAR Point Cloud Projection

Usually, a 3D LiDAR point cloud is represented by Cartesian coordinates (x,y,z) and additional features such as intensity and RGB values. However, processing in dimensions higher than those of image data requires more computational resources. So, Andres et al. [11] proposed a method for converting a 3D LiDAR point cloud into an image using spherical surface mapping. This method of converting a 3D LiDAR point cloud into a 2D range image form facilitates processing of the point cloud and incurs a lower computational cost. Each point (x,y,z) of the data was mapped to an image coordinate (u,v) by using the following equation: (1)$$\left(\begin{array}{c} u\\ v \end{array}\right)=\left(\begin{array}{c} \frac{1}{2}\left[1-\arctan\left(y,x\right)\pi^{-1}\right]w\\ {}[1-\left(\arcsin\left(z,r^{-1}\right)+f_{down}\right)f^{-1}]h \end{array}\right),$$
where (w,h) are the width and height, respectively, and *r* represents $\sqrt{x^{2}+y^{2}+z^{2}}$. *f* is defined as the sensor’s vertical field of view, $f=\left|f_{down}\right|+\left|f_{up}\right|$, where $f_{down}$ and $f_{up}$ represent the LiDAR’s vertical field of view in the downward and upward directions, respectively.

Additionally, the 2D range image can be smoothed as a continuous image [50]. However, the post-process after the conversion of the raw 3D point cloud into a 2D range image may lose many of the details of the original 3D information. Also, that kind of post-processing process can distort the noise more than the original data and may also require additional computational resources. Since this paper aims to determine the traversable region for the raw point cloud, only the described transformation is used. Figure 3 illustrates the conversion of a 3D point cloud into 2D range image using RELLIS-3D. In this study, we converted RELLIS-3D point clouds into range images with class pooling, as mentioned in Section 3.1. In order to use point cloud in a form as similar to image processing as possible, point cloud projection was used as a network input in the form of [w×h×3]. After the spherical projection, three channels consisting of (r,z,i) are used to resemble an RGB image. This utilizes the distance information contained in the LiDAR point cloud but allows it to be used in the same form as image processing.

### 3.3. Network Selection

In deep learning, KD is used to transfer knowledge from cumbersome teacher networks to smaller and simpler student networks [22]. To transfer knowledge, the teacher network must have a high prediction accuracy, and the student network must require less computation than the teacher network. In this study, a model that performs with high prediction accuracy for the RELLIS-3D dataset was applied as the teacher network based on four benchmarking results conducted when the dataset was released. In traversable region detection performed with RELLIS-3D images or point cloud input, the four networks were GSCNN [30], HRNet+OCR [51], KPConv [52], and SaslaNext [25], where GSCNN showed the highest accuracy in an off-road environment. Through this benchmarking, the GSCNN was used as a teacher network to distil the knowledge of the proposed method. The GSCNN is designed for the sharper prediction of object boundaries. It consists of a regular stream, a shape stream responsible for shape processing, and a fusion module that fuses information from the two streams. A regular stream is composed of a backbone architecture that extracts the semantic region features. The shape stream consists of a gated convolution layer (GCL) and local supervision and is designed to process boundary-related information. The results for the two streams were fused using an Atrous Spatial Pyramid Pooling (ASPP) fusion module. In addition, to verify the performance of the GSCNN, the segmentation accuracy was compared using DeepLabV3+ [53] as a baseline. Thus, the GSCNN proved to be an effective network for the clear prediction of object boundaries. Because DeepLabV3+ was initially chosen as a baseline for comparison with GSCNN, this likely means that DeepLabV3+ performs reasonably well on the task at hand and fits the problem domain. Thus, using DeepLabV3+ as a student network yields intuitive results for the proposed distillation method.

### 3.4. Soft-Label Knowledge Distillation (SLKD)

Hinton et al. [22] performed logit distillation using response-based knowledge. The proposed method defines the distillation loss using the pre-trained teacher’s soft label and the student’s soft prediction. In addition, student loss was defined using the student’s hard predictions and the ground truth. Using these two losses, the teacher’s knowledge was successfully transferred to the student. The main goal of the proposed SLKD is to reduce the computational cost of traversable region detection using range images in off-road environments. The range image created by projecting a point cloud has limited heights of 16, 32, 64, 128, etc., owing to the characteristics of the LiDAR sensor. As the number of channels increases, the price of LiDAR sensors also increases, and it is thus difficult to always use LiDAR with a high number of channels. As a result, range images inevitably lack vertical detail compared with regular camera images. Therefore, when using hard predictions for student loss, similar to vanilla knowledge distillation (KD), it is important to consider that the training process may be restricted to capturing detailed vertical information. Considering this, it was expected that performing KD based on soft prediction—which includes continuous probabilities in the range image and considers uncertainty—would yield superior results. Based on this observation, we propose soft-label knowledge distillation (SLKD) for off-road environment segmentation in the form of a range image.

Using the RELLIS-3D range image, we first trained the teacher network using the pre-trained teacher network, and SLKD had the purpose of minimizing the student network’s loss as follows: (2)$$L_{s}=\sum_{i\in input}L_{CE}\left(N_{s}\left(i\right),GT\right),$$
where $L_{CE}$ is the cross-entropy loss, *N* is the network, and GT is the ground truth. While training the student network, the teacher network generated a prediction for the input range image. The distillation loss is defined as follows: (3)$$L_{d}=L_{CE}\left(N_{s}\left(i\right),N_{St}\left(i\right)\right),$$
which is the cross-entropy loss between the teacher network’s soft labels and the student network’s soft predictions. Student loss is defined as follows: (4)$$L_{s}=L_{CE}\left(N_{s}\left(i\right),GT\right).$$

Vanilla KD uses the student loss as the cross-entropy loss of the student network’s hard prediction and ground truth. Using the soft prediction of a student network can affect the backpropagation to obtain a softer probability distribution and less label noise. Total loss is defined as follows: (5)$$L_{total}=w_{1}L_{CE}\left(N_{s}\left(i\right),N_{t}\left(i\right)\right)+w_{2}L_{CE}\left(N_{s}\left(i\right),GT\right),$$
where $w_{1}$ and $w_{2}$ are the user parameters. The total loss supports the student network to follow the teacher network’s distribution and compensates for the weak vertical information of the point cloud range image. An overall pipeline of the proposed method is shown as Figure 4.

## 4. Results

In this section, we describe the implementation details of the proposed method SLKD. We evaluated the proposed method using the RELLIS-3D dataset.

### 4.1. Implementation Details

The proposed method is implemented using PyTorch. For training, we used a resolution of 64×1440 for the point cloud range image. The equipment used included one graphics processing unit (GPU), specifically the NVIDIA RTX A5000. For all the experiments, the total batch size was 16 and the learning rate was $1\times 10^{-6}$. The optimizer used was stochastic gradient descent, with each training run consisting of 100 epochs. In addition, we set user parameter $w_{1}=w_{2}=0.5$. Following conventional segmentation, we adopt the evaluation metric as the mean intersection of union (mIoU) as follows: (6)$$mIoU=\frac{1}{N}\sum_{i=1}^{N}\frac{TP}{TP+FP+FN},$$
where TP,FP,FN represent TruePositive,FalsePositive,FalseNegative. In addition, to evaluate operational efficiency, GFLOPS was used for comparison.

### 4.2. Evaluation

The RELLIS-3D dataset consisted of 13,556 LiDAR scans for the training, testing, and validation of 7800, 2413, and 3343 scans, respectively. Table 2, Table 3 and Table 4 list the experimental outputs obtained using the RELLIS-3D LiDAR scan validation set. For student networks, Kim et al. [54] showed that the MobileNet _v2 [55] encoder-based DeepLabV3+ is the most efficient architecture for several segmentation networks. In addition, to verify the performance of the SLKD, we tested the method with MobileNet_v2_120d and MobileNet_v3 (small and large) [56] encoders. MobileNet is a CNN architecture designed for efficient and lightweight deep learning tasks, particularly on mobile and edge devices. The “120d” in MobileNetv2_120d indicates the width multiplier, which controls the number of channels in each layer. We use for evaluation that having more parameters produces a better output in SLKD. Also, MobileNet_v3_large and small differ in the number of parameters they have.

Table 2 shows IoU per class and mIoU of the validation set. As shown in Table 2, the SLKD performed better than the student network without distillation. The original student network DeepLabV3+ with MobileNet_v2, MobileNet_v2_120d, and MobileNet_v3_large achieved 56.48%, 56.46%, 54.37% mIoU, respectively, whereas applying SLKD achieved mIoU values of 57.28%, 56.75%, and 55.70%, respectively. In particular, when SLKD is applied to MobileNet_v3_large, it improves mIoU by 1.34% compared to before using our method. Moreover, in the case of MobileNet_v2 and MobileNet_v2_120d, IoU of the ‘Sky’ class decreases while the IoU for other classes increases. This indicated that network’s prediction performance is stably normalized with SLKD. In the case of MobileNet_v3_large, IoU increased for every class, and it can be seen that SLKD successfully improved the network’s performance.

The semantic segmentation results of the RELLIS-3D dataset are shown in Figure 5. In Figure 5 and Figure 6, the blue, green, yellow, and pink colors represent the sky, traversable, non-traversable, and obstacle regions, respectively. We can observe that the student network without distillation in Figure 5c predicts the obstacle class comprehensively compared to the teacher or ground truth. However, with the proposed SLKD, Figure 5d shows a more detailed prediction following teacher’s predictions. In addition, by comparing (e) and (f), (g) and (h), and (i) and (j) in Figure 5, SLKD-applied segmentation outputs are closer to the ground truth than those of the original networks. The white boxes in Figure 5 show that the predictions of the non-traversable regions are stable after applying SLKD. For example, Figure 5c–i, in the white box, predict sparsely in the non-traversable region so that this can cause wrong motion planning, leading to collisions or unstable areas. However, using our proposed method, with the same computational cost, the non-traversal area densely appears, as shown in Figure 5d,f,h,j. In addition, we can see that MobileNet_v2 performance is improved more than that of MobileNet_v2_120d. This suggests that having more parameters does not always guarantee a higher receptive capacity for the teacher network’s knowledge.

Additionally, it appears that MobileNet_v3_small exhibits a decrease in mIoU performance, as shown in Table 2. However, the detection of the obstacle class is improved, shown in Figure 5j, which is more similar to the ground truth than that in Figure 5i. In addition, as shown in the white boxes of Figure 6c,d, the student network without SLKD shows the incorrect detection of the obstacle, but when SLKD is applied, the output shows a clear prediction. This shows that SLKD successfully supports the normalization of the prediction. The reason for the decrement of mIoU performance is considered to be the fact that a parameter size that is too small, as shown in Table 3, cannot hold sufficient knowledge from the teacher.

The results show the efficiency of SLKD by increasing the performance with the ratio of GFLOPS to mIoU, which indicates the mIoU performance per one GFLOPS. As shown in Table 4, MobileNet_v2 encoder-based DeepLabV3+ can perform most efficiently when the same computational resources are obtained. In addition, the proposed SLKD achieves a simple, smaller network to mimic the teacher’s knowledge so that it can perform better than the original student network using the same computational cost. According to the results obtained, the SLKD can create an autonomous driving system in an off-road environment to perform traversable-region detection without relying on expensive GPU resources. SLKD is an efficient, cheap, and highly accurate solution for processing point clouds in off-road environments.

## 5. Conclusions

In this paper, we proposed a KD method for the segmentation of off-road view range images called SLKD. The proposed method was evaluated using a well-performing off-road view segmentation network and a smaller network using the off-road environment multimodal dataset RELLIS-3D. Through evaluation, the student network that transferred knowledge from the teacher network exhibited superior performance while incurring the same computational cost as the student network without distillation. Additionally, the robustness of the proposed method was verified using the experimental results of various encoders. In future, we aim to apply the proposed KD method to driving cars in real-world scenarios, rather than solely focusing on the test/validation sets of a dataset.

## Figures and Tables

**Figure 1 sensors-24-00079-f001:**
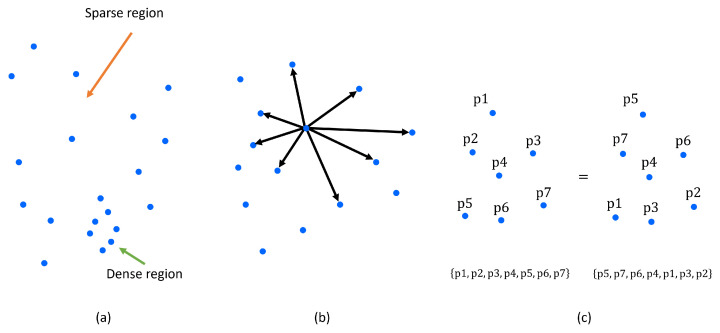
Characteristics of the point cloud data. (**a**) Represents irregularity; (**b**) represents the unstructured nature; (**c**) represents the unordered nature.

**Figure 2 sensors-24-00079-f002:**
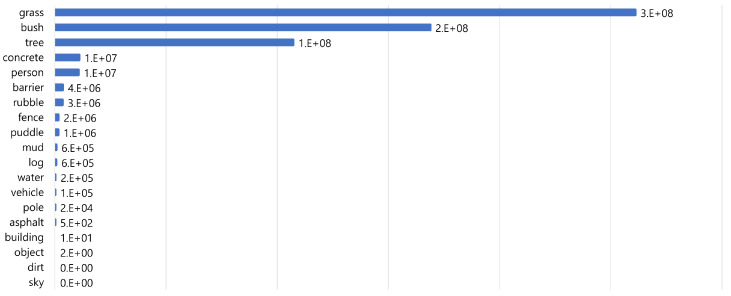
Number of points per class in RELLIS-3D. Grass, bush, tree classes account for about 80% of the total point labels.

**Figure 3 sensors-24-00079-f003:**
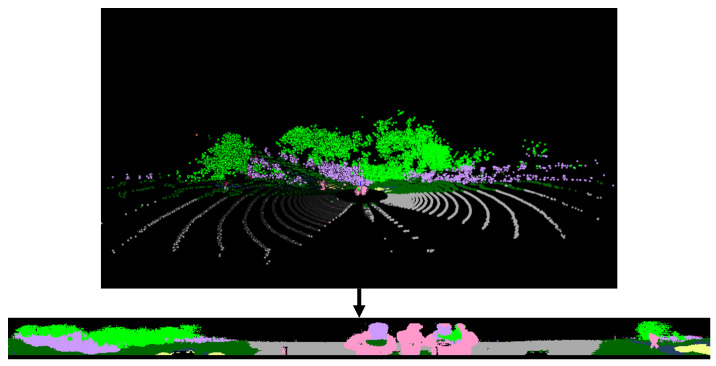
Three-dimensional (3D) point cloud projection to 2D range image. Projection example shows semantic labeled point clouds converted into range images. All points are mapped onto pixels of the range image.

**Figure 4 sensors-24-00079-f004:**
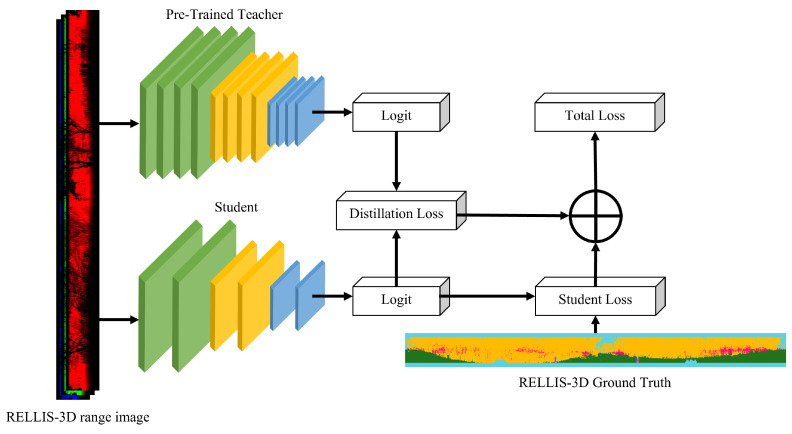
SLKD architecture. Input range image is constructed with 3 channels including (r,i,z). The pre-trained teacher model generates a soft-label and the student generates soft-label prediction, which creates distillation loss. Total loss is defined by the summation of student loss and distillation loss.

**Figure 5 sensors-24-00079-f005:**
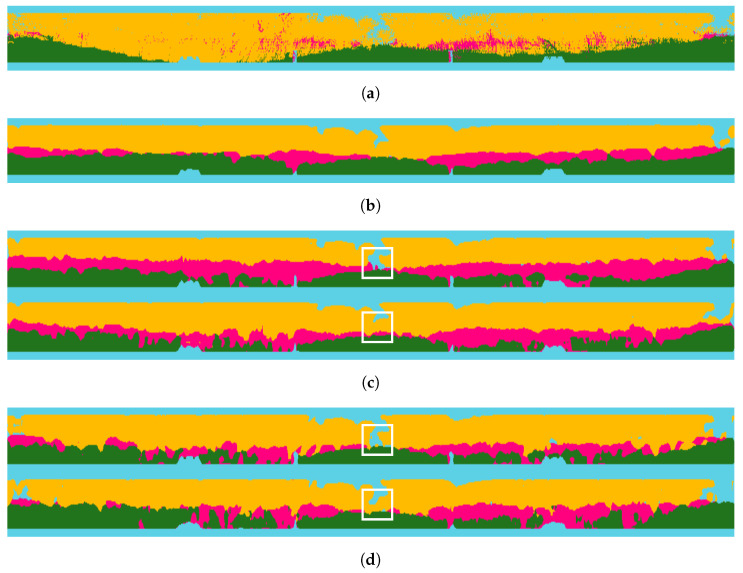
Semantic segmentation results of RELLIS-3D. (**a**) Ground truth; (**b**) teacher (GSCNN); (**c**) DeepLabV3+ with MobileNet_v2 (upper image: w/o SLKD, below image: with SLKD); (**d**) DeepLabV3+ with MobileNet_v2_120d; (**e**) DeepLabV3+ with MobileNet_v3_large; (**f**) DeepLabV3+ with MobileNet_v3_small; In (**c**–**f**), the segmentation results located at the top correspond to the outcomes without the application of the proposed SLKD, while those at the bottom represent the segmented results with SLKD applied. The white boxes show that the predictions of the non-traversable regions are stable after applying SLKD.

**Figure 6 sensors-24-00079-f006:**
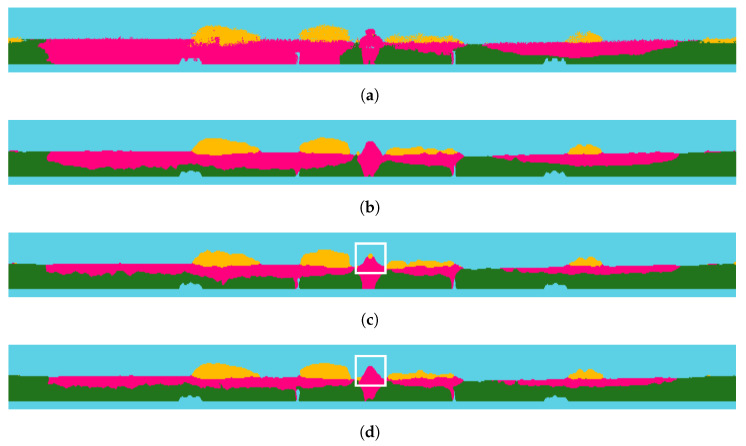
Semantic segmentation results of RELLIS-3D with SLKD of MobileNet_v3_small encoder. (**a**) Ground Truth; (**b**) teacher (GSCNN); (**c**) DeepLabV3+ with MobileNet_v3_small(w/o SLKD); (**d**) DeepLabV3+ with MobileNet_v3_small(with SLKD). The white boxes show that the predictions of the non-traversable regions are stable after applying SLKD.

**Table 1 sensors-24-00079-t001:** RELLIS-3D point clouds class pooling.

Class	Sub-Class
Sky	Sky
traversable region	Grass, Dirt, Asphalt, Concrete, Puddle, Mud
Non-traversable region	Bush, Void, Water, Deep Water
Obstacle	Vehicle, Barrier, Log, Pole, Object, Building, Person, Fence, Tree, Rubble

**Table 2 sensors-24-00079-t002:** Evaluation results of SLKD on RELLIS-3D. The values are in %. Increased value is shown as bold.

Model	Encoder	Sky	Traversable	Non-Traversable	Obstacle	mIoU
Teacher (GSCNN)		95.76	44.50	45.20	53.45	59.73
	MobileNet_v2	95.68	42.00	37.17	51.08	56.48
	MobileNet_v2 (SLKD)	95.55	**42.63**	**37.93**	**53.04**	**57.28**
	MobileNet_v2_120d	95.56	41.44	36.85	52.00	56.46
Student (DeepLabV3+)	MobileNet_v2_120d (SLKD)	95.19	**42.05**	**37.44**	**52.33**	**56.75**
	MobileNet_v3_large	94.59	38.23	34.01	50.60	54.37
	MobileNet_v3_large (SLKD)	**95.56**	**40.20**	**34.37**	**52.67**	**55.70**
	MobileNet_v3_small	94.95	42.28	36.78	51.01	56.25
	MobileNet_v3_small (SLKD)	94.45	39.20	29.49	**54.42**	54.39

**Table 3 sensors-24-00079-t003:** GFLOPS and parameter size of teacher and student networks.

Model	Encoder	mIoU	GFLOPS	Parameter Size
Teacher (GSCNN)		59.73	8493.46	137.27M
	MobileNet_v2	56.48	50.46	2.71M
	MobileNet_v2 (SLKD)	**57.28**		
	MobileNet_v2_120d	56.46	84.99	5.04M
Student (DeepLabV3+)	MobileNet_v2_120d (SLKD)	**56.75**		
	MobileNet_v3_large	54.37	52.13	4.71M
	MobileNet_v3_large (SLKD)	**55.70**		
	MobileNet_v3_small	56.25	32.88	2.16M
	MobileNet_v3_small (SLKD)	54.39		

**Table 4 sensors-24-00079-t004:** Ratio between mIoU and GFLOPS.

Model	Encoder	mIoU/GFLOPS
Teacher (GSCNN)		0.007
	MobileNet_v2	1.119
	MobileNet_v2 (SLKD)	**1.135 (+0.016)**
	MobileNet_v2_120d	0.664
Student (DeepLabV3+)	MobileNet_v2_120d (SLKD)	**0.667 (+0.003)**
	MobileNet_v3_large	1.042
	MobileNet_v3_large (SLKD)	**1.068 (+0.026)**
	MobileNet_v3_small	1.170
	MobileNet_v3_small (SLKD)	1.654

## Data Availability

All data are presented in the paper.

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
