# Peer review of "Knowledge Distillation for Traversable Region Detection of LiDAR Scan in Off-Road Environments"

_sensors, 2023, doi:10.3390/s24010079_

Round 1

Reviewer 1 Report

Comments and Suggestions for Authors

In this paper, the authors propose a knowledge distillation (KD) method for segmenting off-road 1 environment range images. The method is evaluated on the RELLIS-3D off-road environment dataset. In general the paper is well written and technically sound. The following comments should be considered in order to improve the paper. 

1. Elaborate on the transformation from 3D point cloud to 2D image. Post-projection, the resulting 2D image is inherently discrete due to the discrete nature of 3D points. Is there an additional step in the transformation process to achieve a continuous 2D image?

2. Refine the literature review by incorporating relevant sources. Notably, consider citing the following key papers on LiDAR-based detection: DOI: 10.1109/LRA.2023.3309575 and DOI: 10.1109/IROS.2014.6943141.

3. Enhance the introduction section to succinctly encapsulate the paper's contributions. A more concise summary will provide readers with a clearer understanding of the paper's unique value.

Comments on the Quality of English Language

Typically good. 

Author Response

Hello, we have attached the response to your review as a pdf file.
Thanks for the review. -Kind regards, authors

Reviewer 2 Report

Comments and Suggestions for Authors

The manuscript is devoted to the problem of increasing the efficiency of semantic segmentation when processing of a 3D point cloud captured by LiDAR. This task is of great importance for unmanned driving in off-road environment. The authors propose to reduce computational cost by applying the Knowledge Distillation technique. The results presented in the manuscript demonstrate good performance of the proposed solution. I have the following remarks to the manuscript.

The contribution of the manuscript should be clearly and explicitly stated in the Introduction. The results should quantitatively confirm the achievement of the goal. In Conclusion the contributions should be highlighted based on results presented in the paper. In my opinion this is very vague in the manuscript.

Captions for figures 5 and 6 could be improved. I recommend that you clearly state where the proposed SLKD is used. This will make the advantages of your method more visible.

It is inconvenient that many Figures and Tables are located far from the text where they are referenced. It would be better to improve the layout of the manuscript.

The section 2.1 is one paragraph of one page long. To make it easier to read, I recommend breaking it down into smaller paragraphs based on meaning.

Comments on the Quality of English Language

Some minor correction of English is required. For example, the sentence starting on line 276 has error; in line 103: “a task” -> “as a task”.

Author Response

(The authors gave the same response as above.)

Round 2

Reviewer 1 Report

Comments and Suggestions for Authors

The revision is satisfactory.